# The Structural Interactions of Molecular and Fibrillar Collagen Type I with Fibronectin and Its Role in the Regulation of Mesenchymal Stem Cell Morphology and Functional Activity

**DOI:** 10.3390/ijms232012577

**Published:** 2022-10-20

**Authors:** Yuliya Nashchekina, Pavel Nikonov, Nikita Prasolov, Maksim Sulatsky, Alina Chabina, Alexey Nashchekin

**Affiliations:** 1Institute of Cytology of the Russian Academy of Sciences, Center of Cell Technologies, Tikhoretsky pr. 4, St. Petersburg 194064, Russia; 2Laboratory «Characterization of Materials and Structures of Solid State Electronics», Ioffe Institute, Polytekhnicheskaya Str. 26, St. Petersburg 194021, Russia

**Keywords:** molecular and fibrillar collagens, fibronectin, actin cytoskeleton, cell focal contacts

## Abstract

The observed differences in the structure of native tissue and tissue formed in vitro cause the loss of functional activity of cells cultured in vitro. The lack of fundamental knowledge about the protein mechanism interactions limits the ability to effectively create in vitro native tissue. Collagen is able to spontaneously assemble into fibrils in vitro, but in vivo, other proteins, for example fibronectin, have a noticeable effect on this process. The molecular or fibrillar structure of collagen plays an equally important role. Therefore, we studied the interaction of the molecular and fibrillar structure of collagen with fibronectin. Atomic force and transmission electron microscopy showed that the presence of fibronectin does not affect the native structure and diameter of collagen fibrils. Confocal microscopy demonstrated that the collagen structure affects the cell morphology. Cells are better spread on molecular collagen compared with cells cultured on fibrillar collagen. Fibronectin promotes the formation of a large number of focal contacts, while in combination with collagen of both forms, its effect is leveled. Thus, understanding the mechanisms of the relationship between the protein structure and composition will effectively manage the creation in vitro of a new tissue with native properties.

## 1. Introduction

The tissue extracellular matrix (ECM) is as important for life as the cells that it supports [1]. ECM regulates cell behavior including cell survival, proliferation, differentiation, etc. Collagens are also found in the extracellular matrix and play a structuring role in tissue organization. Being fibrillogenic proteins, collagens regulate the interaction of cells and ECM. Collagen I is the most abundant type in human body tissues. The arrangement of collagen molecules in self-assembled fibrils is highly important for the functioning of cells that adhere to them [2]. Cells bind to surrounding extracellular matrix through transmembrane integrin protein complexes, binding through interaction with the structural sites of the cell cytoskeleton, collagen fibrils and other proteins of the ECM [3].

The fibrils are of broad biomedical importance as they play central roles in embryogenesis, tissue repair and pathologies (arthritis, fibrosis, tumor invasion, and cardiovascular disease etc.). Yet there is limited information available to explain the molecular interactions that guide the transition from discrete collagen molecules to tissue [4,5]. It was demonstrated that collagen can be independently guided (i.e., in the absence of other biological molecules) into highly organized structures in vitro [6]. Collagen I spontaneously form fibrils when pH and temperature change [7]. Collagen fibrillogenesis in vitro is a self-assembly process, while the situation in vivo is much more complex. The formation of fibrillar collagen I in vivo occurs in the presence and under the influence of a huge amount of surrounding components of the ECM: fibronectin, fibronectin-binding and collagen-binding integrins and other types of collagens. One of the important features of collagen is its ability to form fibrils even in the absence of cells. Kadler with coauthors demonstrated sufficient initiation of collagen fibrillogenesis in vitro by the BMP-1/tolloid family of metalloproteinases [8]. There are more than 50 known binding partners of fibrillar collagens in vivo [9]. Most of the studies have examined the effect of multiple proteins of ECM on cell function. It has been repeatedly demonstrated that collagen promotes the adhesion and proliferation of cells of various tissue origin [10,11]. Additionally, we and other authors previously demonstrated the effect of collagen on enhancing the osteogenic potential of mesenchymal stromal cells [12,13]. Mesenchymal stromal cells cultured on collagen are able to deposit calcium salts to a greater extent than cells cultured on the surface of culture plastic or other proteins, such as fibrin. Understanding the mechanisms of interaction of the extracellular matrix components with each other and together with cells in vitro will allow us to develop biomimetic approaches for more physiologically relevant tissue production.

Fibronectin is another key tissue component. It is an ECM protein that, binding integrin receptors of the cell surface, acts as a key player for the contact between extracellular environments [14]. The investigation of fibronectin interaction with other ECM component and cells allowed one to better understand the mechanisms of tissue organization in vivo [15,16]. Fibronectin synthesis disruption by cells prevents the formation of a collagen network in vitro and is embryonically lethal in vivo [17,18]. Fibronectin also regulates cell behavior by having adhesive motifs that promote cell adhesion and spread. For example, fibronectin leads to an evident promotion of cell adhesion and spreading in terms of degree and speed [19]. Fibronectin interacts with the α1 chain of collagen I via a gelatin-binding domain [20,21]. Despite the rather long study of the synthesis of extracellular matrix proteins in vitro and in vivo, the understanding of the molecular mechanisms of the interaction between collagen and fibronectin and their impact on the formation and structuring of the extracellular matrix is still insufficiently studied [22].

So back in 1978, Vaheri with coauthors studied the co-distribution of collagen and fibronectin fibrils in the ECM of cultured WI-38 fibroblasts [23]. They observed that the co-localization of collagen and fibronectin molecules began intracellular and persisted into the pericellular space. Thus, it can be assumed that not only fibrillar complexes based on collagen and fibronectin affect the functional activity of surrounding cells in native tissue but also molecular complexes of both, entering the extracellular space, can regulate cell activity. Currently, there are no data on a comparative study of the effects of molecular and fibrillar complexes based on collagen type I and fibronectin on cell activity. Such studies are not only of fundamental interest for understanding the processes of new tissue formation in vivo during the interaction of cells with molecular and fibrillar complexes based on collagen and fibronectin but also of practical interest. When cultivating cells in vitro, collagen is used both in fibrillar form as part of various scaffolds and in molecular form when coating polymeric surfaces, including culture dishes [24].

The aim of this work was to study the effects of the structural features of molecular and fibrillar complexes based on collagen type I and fibronectin on the functional activity of mesenchymal stromal cells. We proposed the investigation of composite proteins and cells and supposed that any inherent molecular-level interactions contribute to tissue genesis. A combination of spectroscopy and microscopy methods was applied in this work to interrogate the structural changes in ECM proteins and functional activities of cells and their mutual influence on each other.

## 2. Results and Discussion

### 2.1. Atomic Force Microscopy (AFM)

One of the main structural and functional components of the extracellular matrix is collagen type I that is in the tissues in the form of fibrils. However, in vitro, it can be used both in fibrillar form, for example, in the form of gels, and in molecular form by coating the surface of culture dishes. Despite the large number of works on the study of collagen, comparative data on the assessment of the effect of its structural forms on cells are still not enough. Therefore, in this work, we take into account the fact that before the formation of fibrils, collagen molecules entering the extracellular space will affect the activity of cells.

It is well known that fibronectin binds to collagen fractions, individual alpha chains, and immobilized helical collagen [25]. Fibronectin is a major extracellular connective tissue component that specifically binds to collagen [26]. The ability of collagen–fibronectin complexes to mediate cellular adhesion has important biological implications in both embryogenesis and tumor growth and spread.

We formed molecular and fibrillar collagen type I extracted from the tendons of rat tails. Composite collagen films were formed with fibronectin. The structure of the resulting surfaces was characterized using AFM (Figure 1).

In concordance with Mercier et al., our results confirm that with the application of a collagen solution in such a concentration to the surface of a cover slip, collagen molecules do not assemble into fibrils but remain in molecular form [24,27]. The AFM method demonstrated the absence of collagen fibrils under the given conditions of protein application on samples (Figure 1a,e). Low concentrations of fibronectin in the solution also make it possible to apply the protein in molecular form, as evidenced by the absence of fibronectin fibrils on the surface of the sample (Figure 1c). Under the optimal conditions for the self-assembly of collagen molecules, namely, sufficiently high protein concentration, pH and ionic strength, native fibrils are formed in vitro (Figure 1b,f). The structure of fibrils obtained in this way was already analyzed in our previous studies and proved to be native [28]. A fibril contains an overlap or high-electron density regions (about 0.46 D long) where the side-by-side overlapping of adjacent triple helices occurs as well as a gap or low-electron density regions (about 0.54 D long) with some space between the ends of the longitudinally lined-up molecules.

When analyzing a fibrillar collagen formed in the presence of fibronectin, inclusions of a spherical shape were found with the diameter equal to the collagen fibrils’ diameter as large as several nanometers. Spherical inclusions are evenly distributed over all sample surfaces based on fibrillar collagen. Preliminary data showed that with a decrease in the content of fibronectin in a collagen solution, the number of spherical inclusions on the surface of fibrillar collagen decreases (unpublished data). Based on the obtained results, we made an assumption that this method of collagen fibril formation with the introduction of this amount of fibronectin would organize a separate phase in the form of spheres with a diameter of several nanometers.

It is well known that collagen rapidly assembles under physiological conditions in vitro, whereas its network formation is strongly dependent on cell activity and the presence on fibronectin in vivo [18,29,30]. Recently investigated the molecular interactions between collagen and fibronectin by using experimental assays designed to selectively target how the presence of fibronectin influences the assembly of the collagen in a cellular environment [31]. Though it was widely hypothesized that collagen-mediated fibronectin unfolding was responsible for fibronectin fibril growth away from the cell surface [22,32], the adjustment to physiological temperature dominated the unfolding observed in Paten’s study and not fibronectin interaction with collagen. The authors reported that despite displaying a low binding affinity for collagen, fibronectin appeared to stabilize collagen nucleation and accelerate the onset of fibril growth. Another group of researchers showed that fibronectin displays a 10-fold decrease in affinity for the collagen alpha chain above 30 °C [33]. This weak attraction may be functionally beneficial to the growth phase, such that collagen monomers or oligomers can easily displace fibronectin to generate properly banded fibrils [34].

### 2.2. Transmission Electron Microscopy (TEM)

Indeed, our TEM results also confirm Speranza’s data. Figure 2 shows comparison images of collagen in fibrillar form (Figure 2a), as well as with the addition of fibronectin (Figure 2b). It can be seen that a well-structured native form of collagen is formed in both cases. The majority of fibrils were similar in diameter and had the characteristic D-periodic banding pattern. After analyzing the TEM images in the ImageJ program, we concluded that there are no statistically significant differences in the D period of fibrillar collagen (64 ± 10 nm) and fibrillar collagen/FN (62 ± 11 nm). The structure of the fibril has a certain periodicity of light and dark rings. In TEM imaging, we can visualize the bright and dark gap regions. This color gradient is due to the peculiarities of the molecular structure of collagen fibrils. In addition, the diameter of the formed collagen fibrils also does not depend on the presence of fibronectin in the collagen solution during its self-assembly in vitro.

### 2.3. Fourier Transform Infrared Spectroscopy (FTIR)

To gain insight into the fibrillar arrangement of collagen in presence fibronectin, we used Fourier transform infrared spectroscopy to help clarify whether collagen fibrils were formed (Figure 3). According to previously published data, there is an intensity increase at the amide III peak (1.242 cm^−1^) when collagen transitions from monomeric solution to fibrillar structures [35]. Earlier, we also confirmed the possibility of using FTIR spectroscopy to analyze the transition of collagen from molecular to fibrillar form [36]. This peak (1.242 cm^−1^) corresponds to an N-H bending mode that is attributed to the proline side chains in fibrillar collagen [37].

Based on the results obtained, it can be concluded that the presence of fibronectin in the collagen solution does not noticeably affect the final structure of the formed fibrils. In this work, we did not study the kinetics of type I collagen fibrillation in the presence of fibronectin. Such results have already been reported by other researchers. Both in vivo and in vitro, fibronectin has been shown to play a key role in initiating the self-assembly of collagen fibrils [38]; specifically, it increases the number of collagen nucleation sites, resulting in the formation of more collagen fibrils [39]. Careful studies of composite collagen and fibronectin matrices in vitro have demonstrated that fibronectin binds collagen mainly under low tension, and as the matrix matures, two independent networks form, and the collagen matrix becomes the primary tension-bearing element in the mature composite ECM [38,40].

### 2.4. Cell Interaction with Collagen—Fibronectin Surface

#### 2.4.1. Scanning Electron Microscopy (SEM) Morphological Analysis

The understanding of the role of the collagen and fibronectin in cell function has changed dramatically in recent decades. For a long time before the discovery of integrins, the main structural elements of the ECM, collagen and fibronectin, were solely regarded as a structural support. When integrins were identified and it became clear that these cell surface receptors could signal, these views changed. Integrins combine information from the extracellular microenvironment into most intracellular signaling pathways [41,42,43]. Prior to the identification of the collagen-binding integrins, the general view was that cells interact indirectly with collagens via collagen-integrin bridging molecules (COLINBRIs) such as fibronectin [44,45]. According to literature data, fibronectin alone is sufficient to induce the highly efficient spreading of many mammalian cell types in vitro [46]. An important functional unit of fibronectin is its RGD tripeptide motif, which acts in synergy with a PHSRN sequence for binding to integrins. In particular, the RGD motif is crucial for mediating cell adhesion and spreading [47].

Much of the initial confusion around how cells interact with collagen was related to different cell types having different integrin repertoires with different specificities [40]. Most importantly, when studying cell–collagen interactions in relation to cell–fibronectin interactions, both fibronectin and collagen receptors need to be characterized. In this work, we studied an interaction cell line of mesenchymal stromal cells isolated from Wharton’s jelly of the human umbilical cord with collagen/fibronectin substrate. Cell morphology (filopodia presence and cell circularity) is a response to topographical features of the substrate surface, and how the cells adhere and spread on the surface influences their behavior. First, we examined the morphology of the MSCWJ-1 cell line with substrates based on molecular and fibrillar collagen in the absence and presence of fibronectin (Figure 4). As can be seen in the figure, after 1 day of cultivation, the cells in all samples are well spread out. It should be noted that cells on molecular collagen (Figure 4b,e) have a more elongated filamentous shape compared with cells in the control sample (Figure 4a) (cover slip), cells on fibronectin (Figure 4d) or fibrillar collagen (Figure 4c,f).

According to SEM data, no significant difference in the morphology of cells cultured on molecular collagen without fibronectin and with fibronectin was observed, and there is no difference in the morphology of cells cultured on fibrillar collagen with fibronectin and without. Obviously, in the combination of collagen and fibronectin, the main factor determining cell morphology is collagen, which levels the effect of fibronectin. Similar effects were also described earlier [48]. It was demonstrated that native collagen in the fibrillar form inhibited cell spreading on a fibronectin substrate, even though cells had been permitted to form initial cell–substrate attachments. Collagenous proteins were found to inhibit cell spreading on fibronectin noncompetitively. In contrast, collagen did not appear to inhibit the binding of fibronectin to the cell surface or the initial adhesion of cells to a fibronectin substrate. Nagata with coauthors state that the extracellular protein collagen can modulate the interaction of cells with a second matrix protein, fibronectin, by a noncompetitive mechanism. That is, one extracellular matrix protein can modulate the function of another. It is conceivable that several types of extracellular molecules could modulate the behavior of a cell simultaneously or successively. One speculation on an in vivo role for the effect of collagen on fibronectin function is that the very high concentrations of collagen in dense connective tissue might prevent cellular responses to fibronectin. The proportions of collagen and fibronectin in vivo may be different. For example, in a wound, in the acute phase, fibroblasts synthesize more fibronectin, and collagen type I is the main protein of the extracellular matrix after healing [48].

Cells cultured on fibronectin have a distinct morphology (Figure 4d). The cell surface is covered by microvilli, and the development of the filopodia at the borders of the cell is present when the cells are cultured on fibronectin. Since such a large number of filopodia are observed on cells cultured on fibronectin, a large number of focal contacts should also be expected. To confirm this hypothesis, antibodies against vinculin were applied to the cells after 1 day of cultivation.

#### 2.4.2. Focal Adhesion Contact Analysis by Confocal Fluorescence Microscopy

Focal adhesions, the main hub for cell mechanosensing, act as a bridge between the integrin–ECM connection and the cytoskeleton [49,50]. In order to analyze the effects of collagen structure and fibronectin on the cellular distribution of adhesion marker, we observed by immunofluorescence that cells MSCWJ-1 showed a localization of vinculin at the all cells on both molecular collagen and fibrillar collagen with and without fibronectin (Figure 5). Focal contacts on the cell surface were also quantified (Figure 5). The greatest number of focal contacts was visualized in cells cultured on the surface of fibronectin. Indeed, as mentioned above, the presence of the RGD sequence responsible for cell adhesion to a protein substrate in fibronectin ensures a high adhesive ability and, consequently, an increase in the number of focal contacts in cells cultured on fibronectin.

It should be noted that in cells cultured on fibronectin, focal contacts were evenly distributed over the entire cell surface. For cells cultured on molecular collagen and in the control sample, a gradient distribution of focal contacts was observed. In the center of the cell, they were little visualized, and their main amount was concentrated in the near-membrane region. There were no significant differences in the number of focal contacts in cells cultured on molecular and fibrillar collagens in the presence and absence of fibronectin. Obviously, this is also due to the fact that the action of fibronectin in the presence of collagen is reduced. However, in this experiment, a change in the morphology of cells on molecular collagen in the presence of fibronectin can be noted. On molecular collagen, the cells have an elongated filamentous shape, while with the addition of fibronectin, the cell area increased. Apparently, this is due to the fact that the ratio between the amount of collagen and fibronectin on the substrate of molecular collagen is less compared with the substrate based on fibrillar collagen. For a more detailed study of the influence of the degree of structural forms of collagen and fibronectin on the degree of cell spreading, the MSCWJ-1 actin cytoskeleton was analyzed.

#### 2.4.3. Spreading Analysis by Confocal Fluorescence Microscopy

As has been repeatedly noted, the degree of cell spreading is an important parameter of the cellular response to the nature of the substrate. To assess the degree of cell spreading and organization of the actin cytoskeleton, cells after 1 day of cultivation on the analyzed substrates were stained with rhodamine phalloidin. To analyze the collagen structure, the protein was stained with immunofluorescent antibodies to native type I collagen. As seen in Figure 6h,r collagen molecules organized into fibrillar structures that evenly covered the entire surface. The fibril diameter corresponds to the diameter of native collagen fibrils. The molecular structure of collagen is not stained by antibodies to type I collagen.

After staining the cells with rhodamine phalloidin, the morphology results from the analysis of the actin cytoskeleton confirmed the results obtained after staining the cells with antibodies to vinculin. Cells plated on all substrates for 1 day spread well and formed an extensive array of parallel actin stress fibers (Figure 6). Cells cultured on molecular collagen had the most elongated shape (Figure 6d). When analyzing the area of cell spreading, a significant increase in the area of cells cultured on fibronectin was revealed. However, in combination with molecular and fibrillar collagen, the presence of fibronectin, on the contrary, helped to reduce the degree of cell spreading.

When analyzing the results of confocal microscopy, a green fluorescent glow was found in the perinuclear region in cells cultured only on molecular collagen. Moreover, the presence of fibronectin did not affect the manifestation of fluorescent luminescence. We assume that this glow is due to the presence of collagen inside the cells, which was specifically stained with antibodies to type I collagen. Apparently, the molecular structure of collagen promotes the synthesis of type I collagen by cells.

## 3. Materials and Methods

### 3.1. Collagen Surface Formation

Collagen I was isolated from rat tail tendons by acetic acid extraction. Molecular collagen was formed from collagen solution with concentration 0.1 mg/mL in 0.1% acetic acid when applied to cover slip [24,27]. Then the collagen was air-dried for 1 day at room temperature. To form a composite surface of molecular collagen, an aqueous solution of fibronectin bovine protein (Plasma, Gibco, Waltham, MA, USA) at a concentration of 0.05 mg/mL was added to the protein solution.

Fibrillar collagen was formed by dissolving of a protein in 0.01% solution of acetic acid (Reactiv, Saint-Petersburg, Russia) to concentration 2 mg/mL. For the fibrils’ formation, 1 M of KH_2_PO_4_ (Reactiv, Saint-Petersburg, Russia) was added to collagen solution to final salt concentration of 20 mM and applied to a cover slip [24,27]. A solution of collagen with salt was kept for 15 min in ammonia atmosphere. After the incubation, the samples were washed twice with phosphate-buffered saline (PBS) (Biolot, Saint-Petersburg, Russia).

Fibrillar collagen surface with fibronectin was formed after adding fibronectin solution with concentration 0.05 mg/mL to collagen solution with concentration 2 mg/mL. For the fibril formation solution, 1 M of KH_2_PO_4_ was added to collagen/fibronectin solution, and then this solution with salt was kept for 15 min in ammonia atmosphere were washed with PBS.

### 3.2. Atomic Force Microscopy

Structure of molecular and fibrillar collagen with/without fibronectin was evaluated using a Dimension 3100 (Veeco, Plainview, NY, USA) atomic force microscope.

### 3.3. Transmission Electron Microscopy

For ultrastructural analysis by transmission electron microscopy, 10 μL samples containing collagen fibrils were transferred onto 200-mesh Formvar-supported and carbon-coated copper grids (Ted Pella, Inc., Redding, CA, USA). Collagen fibrils deposited on the grids were stained negatively with 1% phosphotungstate (pH 7.0) and visualized with a transmission electron microscope operating at 75 kV (model H-7000, Hitachi Ltd., Tokyo, Japan).

### 3.4. Fourier Transform Infrared

Molecular and fibrillar collagen samples with/without fibronectin were analyzed using a FTIR spectrometer IR Prestige-21 (Shimadzu, Tokyo, Japan) in reflection mode, in the 650–5500 cm^−1^ range and with spectral resolution 2 cm^−1^.

### 3.5. Cell Cultivation

In vitro biocompatibility was examined using the cell line of mesenchymal stromal cells isolated from Wharton’s jelly of the human umbilical cord (MSCWJ-1). The cell lines were obtained from the Vertebrate Cell Culture Collection (Institute of cytology RAS, St-Petersburg, Russia). The cells were cultured in polystyrene flasks in DMEM/F12 (Lonza, St. Louis, MO, USA) supplemented with 10% fetal bovine serum (FBS; HyClone, St. Louis, MO, USA), 1% penicillin/streptomycin (Sigma-Aldrich, Darmstadt, Germany) at 37 °C in a humidified atmosphere of 5% CO2 in air. Sub-confluent cells were passaged by using trypsin–EDTA (0.25% (*w*/*v*) trypsin, 1 mM EDTA).

### 3.6. Scanning Electron Microscopy

The cell morphology on molecular and fibrillar collagen structures with/without fibronectin after 1 day cultivation was evaluated using a JSM-7001F (Jeol, Tokyo, Japan) scanning electron microscope (SEM). A 30 nm thick layer of gold was deposited by magnetron sputtering on an Emitech K950 setup (Quorum Technologies, Lewes, UK) to ensure charge drain.

### 3.7. Fluorescence Staining of MSCs

Cell line MSCWJ-1 were fluorescence stained by rhodamine phalloidin and anti-vinculin antibodies in order to study the effects of fibronectin influence and collagen structure on MSCWJ-1 adhesion, spreading and the presence of focal contacts. Pure glass was used as a positive control. A precise description of the technique for fluorescent staining of cells was described in our previously published work (Nashchekina et al., 2020). Briefly, staining was performed as follows. After the cultivation period for 1 day, the medium was removed, and the adherent cells MSCWJ-1were washed with PBS, fixed with a 4% formaldehyde solution (Sigma-Aldrich, Saint Louis, MO, USA). Next, a detergent solution was added to the cells.

Rhodamine phalloidin (Thermo Fisher Scientific, Carlsbad, CA, USA) was used to stain the actin and DAPI (ab104139; Abcam, Cambridge, MA, USA) was used to stain the nuclei. The cytoskeleton organization and spreading were analyzed using a confocal microscope Olympus FV3000 (Olympus Corporation, Tokyo, Japan). Focal contacts were study after cell incubation with anti-vinculin antibodies (ab129002; Abcam, Cambridge, MA, USA). Following this incubation, the goat anti-rabbit IgG (H & L chain) antibodies (ab205718; Abcam, Cambridge, MA, USA) and DAPI were added to cells MSCWJ-1. Then, using a confocal microscope Olympus FV3000 (Olympus Corporation, Tokyo, Japan), we analyzed the cells for the presence of focal contacts.

### 3.8. Cell and Focal Contacts Counts

To study the effects of fibronectin influence and collagen structure on cellular adhesion and number of focal contacts, cells MSCWJ-1 were cultured for 1 day. Five different fields on each sample picture were used at a wavelength of 365 nm (DAPI) by a fluorescence microscope Pascal (Carl Zeiss Jena GmbH, Jena, Germany). The ImageJ program was used to count the nuclei in each picture and the number of focal contacts [51].

### 3.9. Statistical Analysis

All experiments were repeated in 3–5 times. An ANOVA and *t*-test were performed using Microsoft Excel software to analyze the statistically significant differences between samples. Data were considered to be statistically important when *p* < 0.05.

## 4. Conclusions

The extracellular matrix that surrounds cells in body tissues is a complex multicomponent system in which all components are interconnected and influence each other. Not only the composition but also the structure of the extracellular matrix play an important role in the formation of tissues in vivo. Understanding the structure, composition and mechanisms of interaction between the components of the extracellular matrix is an important fundamental problem of modern tissue engineering and regenerative medicine when creating tissue-like structures with native properties in vitro.

In this work, the main attention is paid to the study of the influence of the structure of this important protein of the extracellular matrix as collagen on the functional activity of cells in vitro. It has been shown that during the transition of collagen from the molecular to the fibrillar form, the influence of its structure on cultured cells increases: The morphology of the cells changes, and they acquire an elongated fusiform shape more characteristic of mesenchymal cells. The quantitative analysis of the data obtained confirms the decrease in the area of spread cells cultured on fibrillar collagen compared with cells cultured on molecular collagen. At the same time, despite the decrease in the surface area of cultured cells, the degree of cell interaction with the substrate based on fibrillar collagen does not decrease, which confirms the data on the analysis of the vinculin membrane protein and the quantitative assessment of intercellular contacts.

As noted earlier, other components of the extracellular matrix, including fibronectin, also affect the processes of collagen self-assembly. In our study, we showed that a low concentration of fibronectin in a collagen solution (up to 5%) does not affect the shape and size of collagen fibrils formed under physiological conditions in vitro. At the same time, based on the results of atomic force and transmission electron microscopy, it can be assumed that under these conditions, fibronectin forms a separate phase without being integrated into the collagen fibril during its self-assembly in vitro. At the same time, fibronectin has a significant effect on the shape of cells and the number of focal contacts being on the glass surface in the absence of collagen. Additionally, when combined with molecular or fibrillar collagen, the effect of fibronectin on mesenchymal cells in vitro is leveled. Currently, the literature presents a limited amount of data on the mutual influence of the components of the extracellular matrix on each other and on cultured cells.

Thus, based on the results obtained, it can be concluded that the nature of the interaction of collagen and fibronectin, as well as the properties of the resulting composite film, is influenced by the ratio of collagen and fibronectin. It was shown that in a sample with fibrillar collagen where the amount of fibronectin is insignificant, the structure of the formed collagen fibrils does not change. There was no perceptible difference in the properties of cultured cells on collagen samples with fibronectin and without fibronectin. Apparently, collagen rather than fibronectin predominantly determines the structural and chemical properties of such composite films.

In the future, we plan to study in more detail not only qualitative but also quantitative dependences of the influence of extracellular matrix proteins on each other, as well as to evaluate the responses of cells to structural changes in composite protein structures.

## Figures and Tables

**Figure 1 ijms-23-12577-f001:**
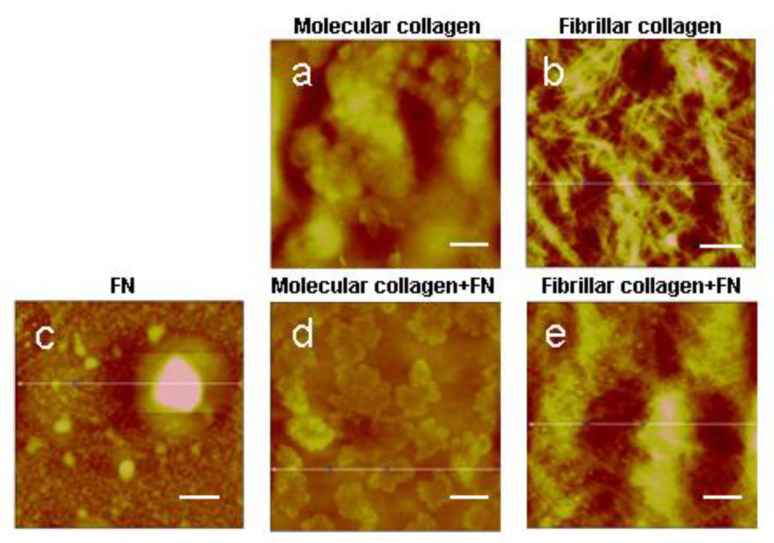
AFM images of molecular and fibrillar collagen with/without fibronectin (FN): (**a**)—molecular collagen; (**b**)—fibrillar collagen; (**c**)—fibronectin; (**d**)—molecular collagen + fibronectin; (**e**)—fibrillar collagen + fibronectin. Scale bar 1 µm.

**Figure 2 ijms-23-12577-f002:**
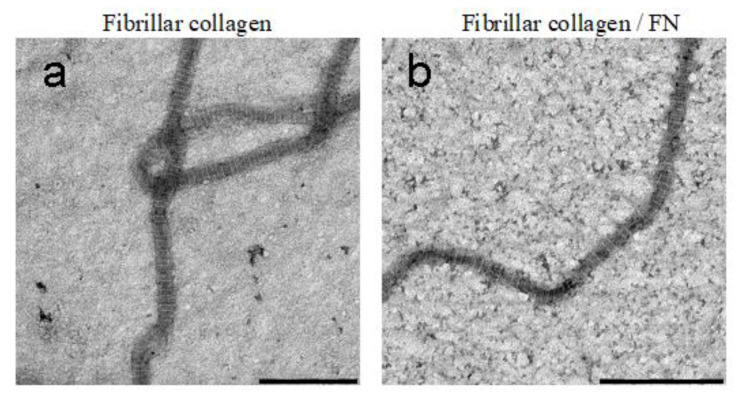
TEM images of fibrillar collagen with/without fibronectin: (**a**)—fibrillar collagen; (**b**)—fibrillar collagen + fibronectin. Scale bar 500 nm.

**Figure 3 ijms-23-12577-f003:**
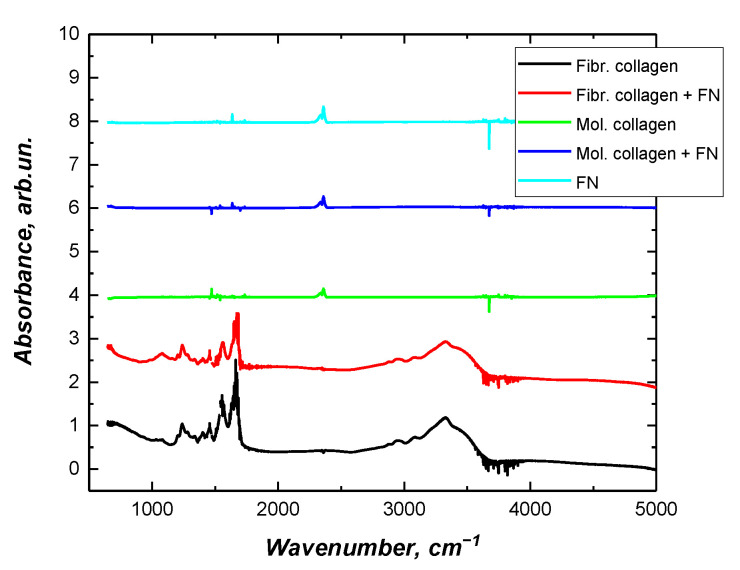
Fourier transform infrared spectra of fibrillar collagen with/without fibronectin.

**Figure 4 ijms-23-12577-f004:**
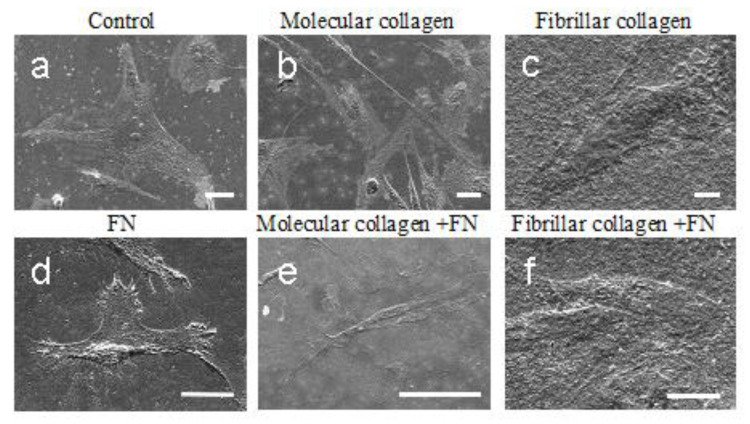
SEM images of MSCWJ-1 cell line after 1 day cultivation on molecular and fibrillar collagen with/without fibronectin (FN): (**a**)—control (cover glass), (**b**)—molecular collagen; (**c**)—fibrillar collagen; (**d**)—fibronectin; (**e**)—molecular collagen + fibronectin; (**f**)—fibrillar collagen + fibronectin. Scale bar 10 µm.

**Figure 5 ijms-23-12577-f005:**
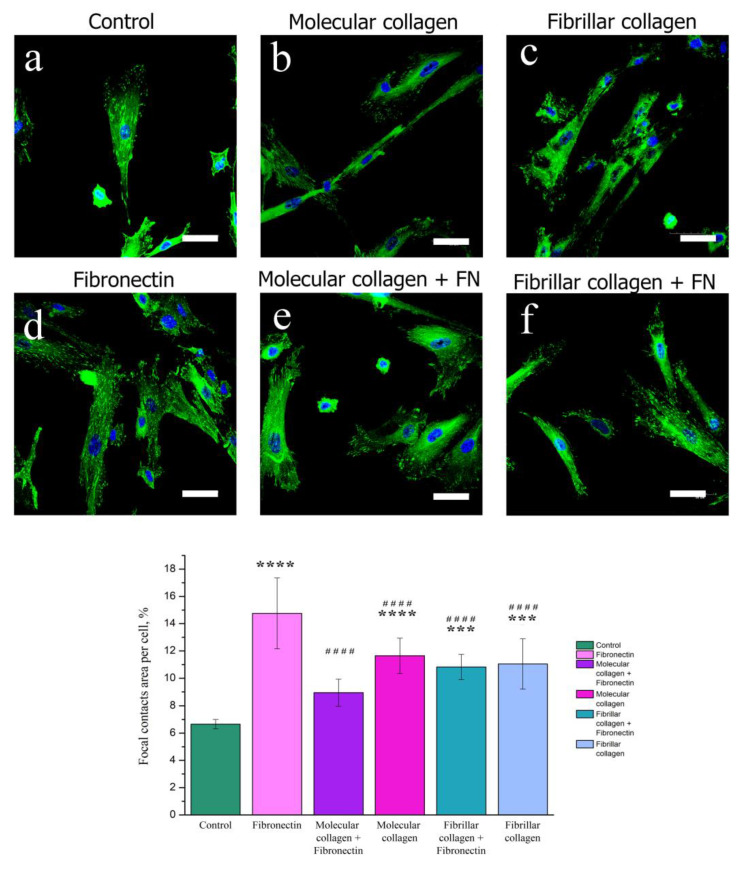
Fluorescence micrographs of MSCWJ-1s with vinculin (green) and nuclei (blue) stained after one days of cultivation on molecular and fibrillar collagen with/without fibronectin (FN): control (cover glass) (**a**), molecular collagen (**b**); fibrillar collagen (**c**); fibronectin (**d**); molecular collagen + fibronectin (**e**); fibrillar collagen + fibronectin (**f**). Statistics of cell adhesion on molecular and fibrillar collagen with/without fibronectin after 1 day cultivation. (n = 7–12: ****—*p* < 0.0001 and ***—*p* < 0.001 compared with the control, ####—*p* < 0.0001 compared with the fibronectin). Scale bar 50 µm.

**Figure 6 ijms-23-12577-f006:**
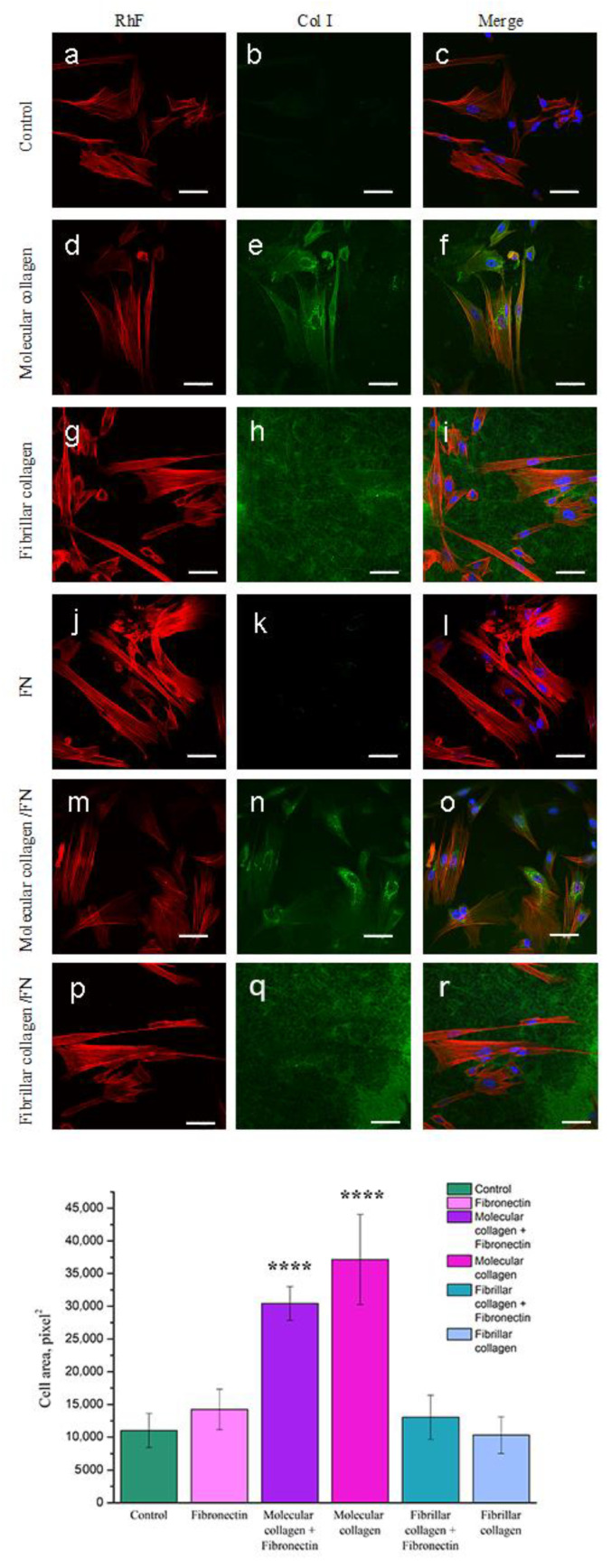
Fluorescence micrographs of MSCWJ-1s with actins (red), collagen fibrils (green) and nuclei (blue) stained after one days of cultivation on molecular and fibrillar collagen with/without fibronectin (FN): control (cover glass) (**a**–**c**), molecular collagen (**d**–**f**); fibrillar collagen (**g**–**i**); fibronectin (**j**–**l**); molecular collagen + fibronectin (**m**–**o**); fibrillar collagen + fibronectin (**p**–**r**). Statistics of cell area (spreading) on molecular and fibrillar collagen with/without fibronectin after 1 day cultivation. (n = 7–21: ****—*p* < 0.0001 compared with the control). Scale bar 50 µm.

## Data Availability

Data available upon request.

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
