# Peer review of "The Structural Interactions of Molecular and Fibrillar Collagen Type I with Fibronectin and Its Role in the Regulation of Mesenchymal Stem Cell Morphology and Functional Activity"

_ijms, 2022, doi:10.3390/ijms232012577_

Round 1
Reviewer 1 Report
This is an admirable attempt to compare the molecular and fibrillar collagens I with fibronectin and their potential role in the extracellular matrix (ECM). Yuliya Nashchekina et al. have used AFM, TEM, and FTIR to study the structural difference between molecular and fibrillar collagens I with fibronectin (FN). The researchers investigated the interaction of cells with collagen–fibronectin surfaces via SEM and confocal fluorescence microscopy. They have paved the way for understanding the role of collagen and fibronectin in ECM. Therefore, I recommended that the paper be published in the International Journal of Molecular Sciences. However, I have some suggestions regarding the manuscript, detailed below, that should be addressed:
Figure 1 shows that FN can bind both molecular and fibrillar collagen. How about collagen and FN concentrations? Although FN bind to collagen, the accessibility to the binding site can differ between molecular collagen and fibrillar collagen. FN-collagen ratio plays a crucial role in network formation.
In the TEM section, the fibrillar collagen and fibrillar collagen/FN form periodic filaments. Is the periodic pattern the same in both cases? A comparison of the periodicity can provide more information.
Minors:
In Figures 5 and 6, it is better to color the bars of different samples using different colors.
Author Response
Response to Reviewer 1 Comments
This is an admirable attempt to compare the molecular and fibrillar collagens I with fibronectin and their potential role in the extracellular matrix (ECM). Yuliya Nashchekina et al. have used AFM, TEM, and FTIR to study the structural difference between molecular and fibrillar collagens I with fibronectin (FN). The researchers investigated the interaction of cells with collagen–fibronectin surfaces via SEM and confocal fluorescence microscopy. They have paved the way for understanding the role of collagen and fibronectin in ECM. Therefore, I recommended that the paper be published in the International Journal of Molecular Sciences.
Dear Reviewer, we are very grateful to You for manuscript review and valuable comments. We tried to take into account all recommendations, expanded and revised the results obtained, which do allowed us to significantly improve the manuscript. All corrections in the text are marked in yellow. Thanks a lot.
However, I have some suggestions regarding the manuscript, detailed below, that should be addressed: Figure 1 shows that FN can bind both molecular and fibrillar collagen. How about collagen and FN concentrations? Although FN bind to collagen, the accessibility to the binding site can differ between molecular collagen and fibrillar collagen. FN-collagen ratio plays a crucial role in network formation.
The Reviewer noted a very important problem of the interaction of extracellular matrix proteins. Indeed, the quantitative ratio of proteins among themselves determines, among other things, the nature of their interaction with each other. Based on the results of many researchers and our results, including, depending on the concentration of the collagen solution applied to the surface of the board, collagen can be deposited in the form of molecules – with a small concentration of protein or in the form of fibrils – when a sufficiently large concentration of collagen is reached. In our studies, we used a solution with a concentration of 0.01 mg/ml for the formation of molecular collagen, and 2 mg/ml for the formation of fibrillar collagen. Also in our study there were samples with pure fibronectin and fibronectin in an amount of 0.05mg/ml added to a collagen solution. Based on the results obtained, we assume that fibrillar collagen is first formed for a collagen solution with a concentration of 2 mg/ml, while fibronectin forms a separate phase. For a collagen solution with a small protein concentration (0.01 mg/ml), the collagen and fibronectin concentrations are very close and there is probably a fairly close interaction of proteins with each other. Therefore, it can be expected that the contribution of fibronectin to the structure and properties of a sample based on molecular collagen is greater than that of a sample based on fibrillar collagen and fibronectin.
In the TEM section, the fibrillar collagen and fibrillar collagen/FN form periodic filaments. Is the periodic pattern the same in both cases? A comparison of the periodicity can provide more information.
After statistical analyzing of the TEM images in the ImageJ program, we can conclude that there are no statistically significant differences in the D period of fibrillar collagen (64±10 nm) and fibrillar collagen/FN (62±11 nm). This data were added to the manuscript.
Minors:
In Figures 5 and 6, it is better to color the bars of different samples using different colors.
We agree that different colors for different bars look more understandable to readers. New drawings are inserted into the corrected version of the manuscript.
Reviewer 2 Report
In the article: “Structural interactions of the molecular and fibrillar collagens I with fibronectin and their role in regulation of mesenchymal stem cell morphology and functional activity” the authors discussed about the interaction of the molecular and fibrillar structure of collagen with fibronectin demonstrating that the collagen structure affects the cell morphology with the aim to manage the creation in vitro of a new tissue with native properties.
Overall, this manuscript results very interesting, the authors clearly explain the rational of the study and discussed the topic point by point.
However, we would like to invite the authors to clarify some minor points:
1. Please check the check punctuation and spaces;
2. Figure 4; please specify the magnification
3. Figure 5: the green appear very strong and the quality of images is not so good, are the single channels (blue and green) available?
4. Figure : the scales bare are not clear; What about the magnification?
5. Figure 5-6: please specify the magnification used to obtain the images;
6. Conclusion; this section appear so long, the authors should better summarize the main point.
Author Response
In the article: “Structural interactions of the molecular and fibrillar collagens I with fibronectin and their role in regulation of mesenchymal stem cell morphology and functional activity” the authors discussed about the interaction of the molecular and fibrillar structure of collagen with fibronectin demonstrating that the collagen structure affects the cell morphology with the aim to manage the creation in vitro of a new tissue with native properties.
Overall, this manuscript results very interesting, the authors clearly explain the rational of the study and discussed the topic point by point.
Dear Reviewer, we are very grateful to you for manuscript review and valuable comments. We tried to take into account all your recommendations, expanded and revised the results obtained, which allowed us to significantly improve the manuscript. All corrections in the text are marked in yellow. Thanks a lot.
However, we would like to invite the authors to clarify some minor points:
- Please check the check punctuation and spaces.
We have corrected punctuation errors and spaces.
- Figure 4; please specify the magnification
In Figure 4, the scale bar for each photo corresponds to 10 microns. Each photo had its own magnification but traditionally the indication of the magnification size is incorrect because when the photographs are scaled, the magnification level also changes. Below we present the original photos for Figure 4.
|
|
||
Figure 4.a - magnification 700, Figure 4.b - magnification 500, Figure 4.c - magnification 2000, Figure 4.d - magnification 1500, Figure 4.e - magnification 500, Figure 4.f - magnification 1500.
- Figure 5: the green appear very strong and the quality of images is not so good, are the single channels (blue and green) available?
We understand that the photos of the cells are quite bright. However, if we remove the intensity of the glow, then, unfortunately, the focal contacts disappear. We have separated the green and blue channels.
- Figure : the scales bare are not clear; What about the magnification?
- Figure 5-6: please specify the magnification used to obtain the images;
Figure 5 and 6 correspond to magnification 50X
- Conclusion; this section appear so long, the authors should better summarize the main point.
We have changed a paragraph summarizing the results to the Conclusion section. Now this section includes the main conclusions of this study in a clearer and more concrete form.
Thus, based on the results obtained, it can be concluded that the nature of the interaction of collagen and fibronectin, as well as the properties of the resulting composite film, is influenced by the ratio of collagen and fibronectin. It was shown that in a sample with fibrillar collagen where the amount of fibronectin is insignificant, the structure of the formed collagen fibrils does not change. There was no perceptible difference in the properties of cultured cells on collagen samples with fibronectin and without fibronectin. Apparently, collagen rather than fibronectin predominantly determines the structural and chemical properties of such composite films.
